# A methodological framework for exploring SME finance with SAFE data

**Marie Finnegan** [1,2‡] *, **Lucía Morales** [2]

**1** School of Business, Atlantic Technological University, Galway, Ireland, **2** School of Accounting, Economics & Finance, Technological University Dublin, Dublin, Ireland

‡ MF is senior author on this work.
* marie.finnegan@atu.ie

**Data Availability Statement:** A description of the data set and the third-party source: The survey on the access to finance of enterprises (SAFE) provides information on the latest developments in the financial situation of enterprises, documents

## Abstract

Research on small and medium-sized enterprises (SMEs) access to bank finance is vital for the euro area economy. SMEs heavily represent the European business sector, employing around 100 million people and accounting for more than half of the Gross Domestic Product. Research studies in the field often rely on the ECB/EC Survey on the Access to Finance of Enterprises (SAFE). Many studies employ probit or logit models with categorical dependent variables derived from SAFE. The research findings show that hardly any study employs the simpler linear probability model (LPM), with a dominant lack of research providing evidence that justifies the model selection process and suitability. However, it is well known that different econometrics models can lack consistency and frequently yield different results. Yet, the literature has no consensus on the best econometric approach. In addition, there is a lack of robustness tests in the literature to ensure model validity, underlining the need for a comprehensive review of the methodological framework that dominates SAFE data use. This paper addresses the identified research gap by introducing a robust methodological framework that helps researchers identify and choose an appropriate categorical model when using SAFE data. The study adds significant value to the extant literature by identifying four criteria that need to be considered when selecting the appropriate model among three common binary dependent models: LPM, probit and logit models. The findings show that the probit model was appropriate is all cases but that the LPM should not be disregarded, as it can be used in two cases: when considering the interaction between monetary policy and debt to assets and monetary policy and innovation. The use of the LPM is justified as a less complex econometric model, allowing for clearer communication of the results. This innovative, robust approach to choosing the appropriate econometric categorical dependent model when employing SAFE data contributes to support policy effectively.

## 1. Introduction

Small and medium sized enterprises (SMEs), comprising 99.8% of firms in the euro area economy, play a crucial role in economic growth and employment. In 2022, they contributed 52% of the total value added (€3.95 billion) and represented 64% of total employment in the

trends in the need for and availability of external financing, and measures firms' expectations about their selling prices, costs, and employment, as well as about euro area inflation. The survey results are broken down by firm size, sector, country, firm age, financial autonomy and ownership. The survey is conducted every quarter as of 2024 Q1 (formerly every six month), three quarters by the ECB covering euro area countries (see results below) and once in cooperation with the European Commission covering all EU countries plus some neighbouring countries. If you would like to access the anonymised SAFE microdata, please fill in and return the confidentiality declaration form available at https://www.ecb.europa.eu/stats/ecb_surveys/safe/html/index.en.html#dd to the survey access team via email: survey.accesstofinance@ecb.europa.eu. Other data is publicly available from the ECB Statistical Data Warehouse, Eurostat and the International Monetary Fund. All relevant data are within the paper and its Supporting Information files.

**Funding:** The author(s) received no specific funding for this work.

**Competing interests:** The authors have declared that no competing interests exist.

European Union's non-financial business sector [2]. Euro area SMEs are especially bank-dependent as they find it difficult to borrow in the corporate bond market or raise capital in the stock market due to their opacity and associated risk [3, 4]. Therefore, euro area banks have an important intermediation role to play in supporting macroeconomic stability given that bank credit is among the crucial determinants for SME survival and growth [5, 6]. SMEs access to bank finance has been a major focus for researchers, given this economic relevance and bank dependence.

Much of the literature investigating SMEs access to bank finance access constructs binary dependent variables using the ECB/EC Survey on the Access to Finance of Enterprises (SAFE). SAFE provides information on the latest developments in the financial situation of enterprises and documents trends in SMEs' demand for and access to bank loans. It is a cross-sectional dataset with only a subset of the respondents to a given wave interviewed in another wave. In general, the literature constructs a categorical dependent variable from the survey's Q7a and Q7b, which asks if SMEs sought a loan and if their request was granted [1, 7–15]. Much of the literature employs nonlinear models such as probit—and, to a lesser extent—logit to estimate their categorical dependent variable. However, the literature lacks consensus and clarity regarding the optimal econometric framework that researchers should consider when using categorical dependent variables constructed from SAFE. Yet, it is well understood that different models can yield different results. Further, there is a lack of a diagnostic framework employed in the literature to ensure that models are robust.

This paper contributes to existing literature by introducing a methodological framework aimed at aiding researchers in determining the preferable categorical dependent model when utilising SAFE data. Such a robust approach is necessary as more robust models inform policy more effectively. This paper considers the three most common models used by researchers: the linear probability model and nonlinear models such as probit and logit. It is an extension study on the recent work by Finnegan and Kapoor [1], which compares their probit model to an LPM and logit model. It suggests that researchers should consider using the LPM if four criteria are met given its superior ease of interpretation. It suggests using nonlinear models such as logit and probit in the absence of these criteria and that the choice between these nonlinear models should be based on which model performs better using a proposed comprehensive diagnostic framework. The remainder of this paper is structured as follows. Section 2 explores the literature surrounding SMEs' access to finance and the methodological and diagnostic approach present in the SAFE literature. Section 3 provides insights into the SAFE dataset. Section 4 presents the methodological framework for choosing one model over another. Section 5 applies this framework to the empirical model and sample used by Finnegan and Kapoor [1]. Section 6 offers some concluding remarks.

## 2. Literature

The literature review examines SMEs' access to bank finance using the SAFE dataset and discrete choice models. SAFE's categorical survey responses, particularly from Q7a and Q7b, are often used to construct dependent variable models for assessing whether SMEs sought a loan and if it was granted. Table 1 shows that nonlinear models, especially probit (57%) and, to a lesser extent, logit (28%), are commonly employed in the SAFE literature—since the first article by Artola and Genre [16] which used SAFE—which had started in 2009. The research findings identified a common trend where the linear probability model is rarely used without justifying this econometric model's neglect (7.4%).

Moreover, the literature has no clarity regarding the methodological framework that leads towards more consistent and reliable estimations. Yet, in the econometric field, it is well

**Table 1. Diagnostic review of SAFE literature.**

| Authors | Q7a or Q7b Binary Dependent variable | Methodology | Goodness of fit | Joint significance | Obs |
|---|---|---|---|---|---|
| Artola & Genre [16] | Credit constrained, Q7a,b | Binary, multinominal probit | pseudo-$R^2$ | Wald test | 16,273 |
| Holton et al. [23] | Credit constrained, Q7a,b | Binary probit | pseudo-$R^2$ | None | 38,802 |
| Ferrando & Mulier [24] | Credit constrained, Q7a,b | Probit | None | None | 13,291 |
| Mac an Bhaird [25] | Discouraged, Q7a | Logit model | McF pseudo-$R^2$ | Wald, LR ratio | 6,287 |
| Ferrando et al. [7] | Credit constrained, Q7a,b | Probit model | $R^2$ | None | 68,796 |
| Demoussis et al. [26] | Credit rationed, Q7a,b | Binary probit model | PCP, $R^2$, | Wald test | 49,618 |
| Galli et al. [27] | Apply for bank loans, Q7a | Multinominal logit model | pseudo-$R^2$ | None | 60,058 |
| García-Posada Gómez [28] | Credit constrained, Q7a,b | OLS, probit, IV | $R^2$ | Wald test | 19,375 |
| Kaya & Masetti [8] | Credit constrained, Q7a,b | Probit model. DID | pseudo-$R^2$ | None | 45,305 |
| Ferrando, Popov et al. [9] | Credit constrained, Q7a,b | Probit and OLS | $R^2$ | None | 30,040 |
| McQuinn [10] | Credit constrained, Q7a,b | Three stage OLS | $R^2$ | None | 29,210 |
| Guercio et al. [29] | Credit constrained, Q7a,b | Binary logit models | McF pseudo-R2, PCP, AIC, BIC | LR ratio | 39,675 |
| Corbisiero & Faccia [14] | Credit Constrained, Q7a,b | LPM, Ordered probit model | $R^2$ | None | 34,058 |
| McNamara et al. [30] | Credit rationed, Q7b | Probit model | pseudo-$R^2$ | None | 13,967 |
| Ertan et al. [31] | Credit Access, Q7b | DID | Adjusted-$R^2$ | None | 11,000 |
| Beyhaghi et al. [32] | Credit rationed, Q7b | Logit, multinomial logit | pseudo-$R^2$, PCP | Wald test | 10,598 |
| Moro et al. [33] | Credit Access, Q7a,b | Binary probit regressions | None | None | 18,872 |
| Galli et al. [34] | Apply for bank loans, Q7a | Logit model | pseudo-$R^2$ | Wald test | 60,058 |
| Guercio et al. [35] | Credit Access, Q7a,b | Binary and ordered logit | pseudo-$R^2$ | LR ratio | 7,305 |
| Ferrando, Ganoulis et al. [36] | Credit access, Q7a,b | Ordered logit, DID | pseudo-$R^2$ | None | 91,432 |
| Calabrese et al. [12] | Credit Constrained, Q7a,b | Multinominal logit, Probit | pseudo-$R^2$ , PCP, AIC, BIC | Wald Test | 106,576 |
| Mol-Gómez-Vázquez et al. [37] | Discouraged, Q7a | Multlevel methodology | None | Wald, LR ratio | 20,207 |
| Kallandranis & Drakos [38] | Discouraged, Q7a | Logit model | $R^2$, PCP | Wald test | 122,134 |
| Sclip [11] | Credit Access, Q7a,b | Multinomial logit, probit | pseudo-$R^2$ | LR ratio | 21,766 |
| Ferrando & Mulier [15] | Credit Access, Q7a,b | LPM and probit | pseudo-$R^2$ | None | 7,739 |
| Betz & De Santis [13] | Credit constrained, Q7a,b | DID, OLS, IV | None | None | 13,945 |
| Kallandranis et al. [39] | Credit constrained, Q7a,b | Probit model | $R^2$, PCP | Wald test | 122,134 |
| Finnegan & Kapoor [1] | Credit constrained, Q7a,b | Probit model | None | None | 11,319 |

Notes: This table displays goodness of fit and inference tests used in the SAFE literature

documented that the choice of econometric framework can lead to different outcomes, and that some models perform better than others. There is no evidence why the probit model is the dominant methodology employed, with the literature justifying its use solely on the nature of the binary dependent variable [7–9, 11, 12, 14, 15].

Further, logit and probit models are very alike in that they generally yield similar results and have the same asymptotic properties [17, 18]. Therefore, there is no compelling reason to choose one over another [19], and it is often a matter of personal choice for the researcher for binary dependent variable models [17, 18]. In addition, there may be good reasons to employ an LPM given its ease of computation, interpretation and the fact that its estimated effects are often reasonable and in alignment with practice [20–22].

Table 1 below shows that the extant literature on diagnostic statistics is concise while Table 2 shows some common measures of goodness of fit for binary dependent variables proposed in the econometrics literature and documents their scant use in the SAFE literature.

Table 2. Comparative diagnostics econometrics and SAFE literature.

| Goodness of fit | Econometrics literature | SAFE literature |
|---|---|---|
| Pseudo $R^2$- identified | Need to identify pseudo $R^2$ given plethora of existing measures and definitional differences [43, 44, 45] | 7.1% of literature: Mac an Bhaird (25) and Guercio (29)—use McFadden's pseudo R2 |
| PCP (Percentage of correct predictions) | Need to report PCP, if $\hat{p}i \geq 0.5$, set $\hat{Y}i = 1$; otherwise set $\hat{Y}i = 0$ [40, 41, 42, 46] | 17.9% of literature (12, 26, 29, 32, 38, 39) |
| PRE (Percentage reduction in error) | Need to compare PCP to PRE which compares the estimated model to the null model [40, 48, 49] | Never reported |
| ePCP and ePRE | Need to report ePCP and ePRE which deals with arbitrary choice of 0.5: $\hat{p}i \geq 0.5$, set $\hat{Y}i = 1$; otherwise set $\hat{Y}i = 0$ [47, 52, 53] | Never reported |
| ROC | Need to report ROC curve to display sensitivity and 1—specificity for all possible thresholds [54, 55, 56] | Never reported |
| AIC and BIC | Need to report AIC and BIC, penalties for including additional variables [44, 57, 58], also for LPM | 7.1% of literature (12, 29) |
| Inference tests | | |
| Wald test | [40, 45, 59] | 25% of literature (12, 25, 26, 32, 37–39) |
| Likelihood Ratio | [40, 43, 60] | 14.3% of literature (11, 25, 29, 35, 37) |

Notes. This table indicates tests for goodness of fit and inference tests. Another inference test is the Lagrange multiplier test.

The core findings highlight how 50% of SAFE studies use the pseudo $R^2$ as a goodness of fit measure as the coefficient of determination $R^2$ cannot be applied to nonlinear categorical dependent models as a measure for goodness of fit [40–42]. However, Hemmert et al. [43] and Williams [44, 45] argue that reporting unknown pseudo $R^2$ is meaningless given the plethora of existing measures and their definitional differences. However, only Mac an Bhaird [25] and Guercio et al. [35] acknowledge that they employ McFadden's Pseudo $R^2$ and none of the literature comments on these measures as a goodness of fit. 18% of the SAFE literature uses the percentage of correct predictions (PCP)—which uses a cut-off of 0.5 to assign probabilities: if $\hat{p}i \geq 0.5$, set $\hat{Y}i = 1$; otherwise set $\hat{Y}i = 0$ [40–42, 46].

However, according to Herron [47], Menard [48] and Gelman and Hill [49], reporting PCP on its own is pointless, and the estimated model needs to be compared to the null model. However, none of the SAFE literature reports the percentage reduction in error (PRE), which is a measure comparing the predictive success of the estimated model (PCP) to a null model, that is, the proportion of the dependent variable in the model category of the observed data (PMC) [48, 50, 51]. Further, the econometrics literature suggests that there is a need to report the expected ePCP and ePRE–proposed by Herron [47], which deals with the arbitrary choice of 0.5 used in the PCP and PRE [47, 52, 53]. This measure never appears in the SAFE literature. The receiver operating characteristics (ROC) graph is another goodness of fit measure—a technique for visualising, organising and selecting classifiers based on their performance—proposed in the econometrics literature [54–56] but does not feature in the SAFE literature.

Finally, the Bayesian information criterion (BIC) and the Akaike information criterion (AIC) penalise models for adding additional variables—and can be used to assess both LPM and nonlinear models—have become increasingly popular in the broader literature as measures of goodness of fit to distinguish among models [44, 57, 58]. However, AIC and BIC appear in only 7.1% of the SAFE literature.

Table 2 shows common inference tests—to see if the model is significant—proposed in the econometrics literature for nonlinear binary dependent variables models such as the Wald test [45, 48, 59] and the likelihood ratio test [43, 48, 60]. However, Table 2 shows that the SAFE literature is scant on diagnostic testing for joint significance of variables with the Wald test present in 25% of studies and the Likelihood ratio test present in just 14.3% of literature with no discussion on the results of these tests. A common test for the joint significance of variables for the LPM is the F test, however, the literature does not report the F test when LMPs are employed [14, 15].

The reviewed literature underlines the importance of identifying a robust methodological framework which researchers investigating SMEs access to finance using SAFE data could use to choose one binary dependent variable model over another—given the lack of clarity in the SAFE literature for choosing a binary dependent model. Moreover, the probit and logit models are the focus of this study, given their relative dominance in the literature. While probit or logit may be preferred over LPM given the well documented problems of linear models estimating binary variables there is a need to offer evidence on models performance that enable a research-informed process that supports researchers when assessing which model is the best fit for their study. Some initial elements to be considered in the context of the LPM model are that its estimates are not constrained to the unit interval and that Ordinary Least Square (OLS) estimation imposes heteroskedasticity in the case of a binary response variable [21, 53, 61].

Further, the LMP is identified as problematic as it assumes that the $P_i = E(Y = 1|x)$ increases linearly with X; that is, the marginal or incremental effect of X is constant throughout, and this may not be the case with a binary model [20, 62]. This paper explores if these reasons exist to elevate probit and logit models over LPM in the context of SAFE data. In addition, even though there are a number of problems associated with the LPM, and it is employed in only 7.4% of studies, this paper considers this model for a number of reasons. First, the results emerging when OLS when applied to LPM are often similar to results emerging from maximum likelihood applied to a probit or logit model when sample sizes are large, despite the unboundedness problem inherent in a LMP [21, 53, 63]. Indeed, estimated effects and predictions with LPM are often reasonably good in practice [20–22]. Second, while probit and logit both capture the nonlinear nature of the population regression function better than the LPM, they are harder to interpret [22, 64]. This justifies the investigation of LPM as an alternative to nonlinear models such as probit or logit, given its superior simplicity in the interpretation of results [22, 63, 65]. The next section explores the SAFE dataset.

## 3. Data

SAFE provides information on the latest developments in the financial situation of enterprises and documents trends in SMEs demand for and access to bank loans and is published every six months. There have been 29 SAFE waves conducted starting in 2009 after the financial crisis affected the euro area. The firm level SAFE also includes information on firms' responses to questions regarding their characteristics in terms of age, size, sector, turnover ownership status and legal form. In addition, it includes an assessment of the SMEs own view of their credit risk. The SAFE sample includes only non-financial firms and companies are selected randomly from the Dun & Bradstreet business register [66]. All survey-based percentages are weighted

statistics that restore the proportions of the economic weight (in terms of employees) of each class size, economic activity and country [66].

Relevant aspects to be considered when using SAFE relate to the database limitations. For example, SAFE is a cross-sectional dataset with only a subset of the respondents to a given wave interviewed in another wave. This restricts the use of firm-fixed effects that would help to identify omitted variable bias related to firm-specific heterogeneity [9]. Further, the publicly available SAFE is anonymised and does not identify firms or match them to their banks, unlike other data sources such as credit registers used to study SMEs access to finance arising from UMP [67–71]. Finally, the dataset provides mainly qualitative information, which contains subjective responses to survey questions, which may not be supported by balance sheet information [15].

Even so, it is argued that SAFE is a useful dataset for studying SMEs' access to bank finance. First, SMEs are generally bank-dependent [13, 72, 73] and SAFE is a rich data source on SMEs access to bank finance. Second, it includes discouraged borrowers, giving us a broader view of the credit markets than other data sources such as credit registers [9–11, 74]. Third, it is used extensively in the literature that studies SMEs' access to bank finance. Finally, it is a very reliable data set given that it is conducted by the ECB/EC and is used by the ECB to evaluate its monetary policy interventions on SMEs [31, 75]. Indeed, the ECB conducts validity checks to ensure that survey answers are accurate [11]. The next section outlines a methodological framework for choosing between discrete choice models when using SAFE and applies it to an empirical model.

## 4. Methodology

This section expands on the broad literature that uses SAFE to explore SMEs' access to bank finance, as outlined in Table 1. In particular, it in an extension of the study by Finnegan and Kapoor [1]—which studies the impact of unconventional monetary policy on SMEs' access to bank finance—in terms of assessing the methodological framework when using binary dependent variables and SAFE. It employs this paper, given their focus on the post-crisis period from 2014–2019 relative to the other literature examining UMP and SMEs, which tends to focus on the financial and/or sovereign debt crisis from 2008 [9, 11, 13, 31, 76].

Further, it expands on this work given that this paper uniquely uses measures of risk from the firms' point of view to evaluate if recently leveraged firms and risky firms are more credit-constrained in times of expansive UMP–to act as a counterpart to the risk-taking channel of monetary policy. The risk-taking channel describes how UMP can lead to excessive risk-taking [71, 77–79]. They find that UMP may trickle down to SMEs unevenly due to their location, even in a post-crisis environment for recently leveraged SMEs [1]. Section 3.1 summarises this empirical model, and Section 3.2 outlines the methodological framework for choosing one model over another.

### 4.1 Empirical model

Finnegan and Kapoor's [1] use SAFE data to construct a binary dependent variable—credit constrained—and employ a probit model. This dependent variable 'credit constrained' equals 1 if the firm reported to have (i) applied for bank loans in the previous six months but was rejected (Credit Denied) or (ii) applied but received less than 75% of its demand (Rationed) or (iii) refused credit because it was offered at too high a cost (Refused due to high cost) or (iv) not applied because of possible rejection (Discouraged). Alternatively, the variable equals 0 if the firm reported having applied for bank loans in the previous six months and received

everything or 75% and above. Given that the indicator is equal to 1 if the firm is credit-constrained, a negative coefficient indicates that SMEs are less likely to be credit-constrained.

They propose two hypotheses using the following empirical specifications:

Hypothesis 1 (H1): UMP decreases the probability of firms with increased debt-to-assets being credit-constrained.

To test H1, they model Eq 1 as follows:

$$P(CreditConstrained_{i,c,t} = 1|x)$$
$$= \alpha_i + \beta' MP_{c,t-2} + \gamma' DebttoAssets_{i,c,t} * MP_{c,t-2} + \delta' X_{i,c,t} + \theta' Macro_{c,t-2} + \varphi' BankCh_{c,t-2} + \tau_{c,s}$$
$$+ \varepsilon_{i,s,c,t} \tag{1}$$

$MP_{t-2}$ is the one-year lag (equivalent to two survey waves in SAFE) of the logarithm of the assets of individual central bank balance sheets—minus autonomous factors—for stressed countries. Debt-to-assets increased is a categorical variable, which equals 1 if the firm's debt-to-assets increased, and 0 if it remained the same or decreased in the previous six months. For H1, $\gamma'$ is an interaction term and is the main coefficient of interest that captures if firms with increased debt-to-assets decreased their probability of being credit-constrained during UMP. It is expected that UMP for leveraged firms should make accessing bank finance easier via their improved balance sheets and collateral, and this should translate into a reduction in credit constraints [80, 81]. A negative relationship is, therefore, expected between the probability that a firm is credit-constrained and the interaction between increased debt-to-assets ratio and UMP. S1 Table outlines the main variables, their definition and their data source employed in the regressions for H1 and H2.

Hypothesis 2 (H2): UMP reduces the probability of risky firms being credit-constrained

To test H2, they model Eq 2 as follows:

$$P(CreditConstrained_{i,c,t} = 1|x)$$
$$= \alpha_i + \beta' MP_{c,t-2} + \gamma' FrimRisk_{i,c,t} * MP_{c,t-2} + \delta' X_{i,c,t} + \theta' Macro_{c,t-2} + \varphi' BankCh_{c,t-2} + \tau_{c,s}$$
$$+ \varepsilon_{i,s,c,t}, \tag{2}$$

For H2, the interaction between monetary policy and firm risk variables is the main coefficient of interest—$\gamma'$—as it captures the probability of a risky firm being credit-constrained during periods of UMP. $FirmRisk_{i,c,t}$ is modelled categorically, and the measures for firm risk include a future predictor of risk—profit decreased in the previous six months, as well as a selection of subjective measures of risk—the firm's own view if there has been deterioration in credit history, own economic outlook and own capital in the previous six months and finally, an activity-based measure of risk—innovative activity given that such activity is more uncertain and therefore riskier. Given the low-interest rate environment generated by UMP, banks are expected to chase higher yields. This will manifest in lending to riskier firms via the monetary policy risk-taking channel [76, 82]. A negative relationship is, therefore, expected between the probability that a firm is credit-constrained and the interaction between increased firm risk and UMP.

The model controls for confounding factors that might influence loan supply and loan demand, such as firm-level heterogeneity, the stage of the economic cycle and bank characteristics. $X_{i,c,t}$ is a set of firm level covariates to control for firm heterogeneity with subscript $i$, $c$ and $t$ indicating firm, country and time respectively. $Macro_{c,t,-2}$ is a vector of macroeconomic variables to control for the economic cycle. $BankCh_{c,t-2}$ easures banks' balance sheet health indicators at the country-level which impacts credit supply and demand. Time-fixed effects are added when monetary policy is measured at the country level to exclude unobserved variables

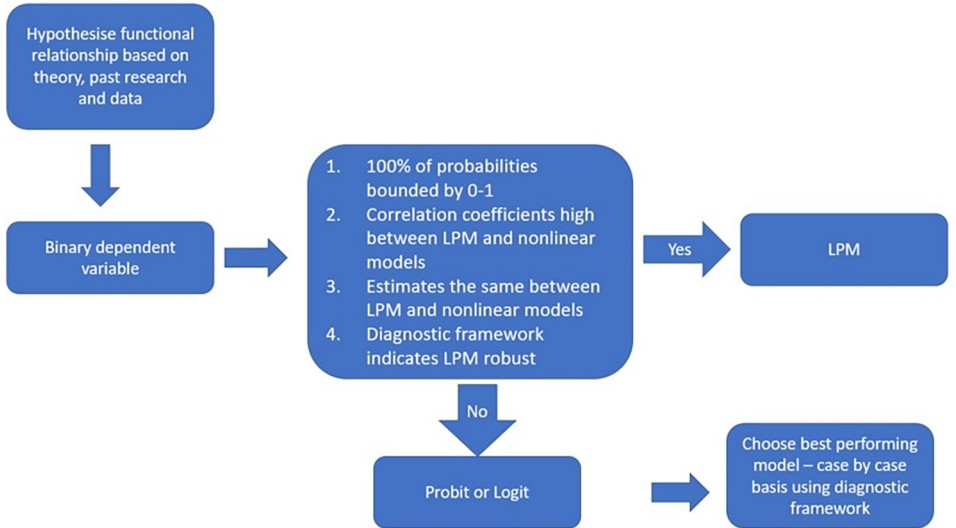

**Fig 1. Framework for choosing binary dependent variable model using SAFE data.** Source: Authors (2024).

that evolve over time but are constant across firms. Further, sector-country fixed effects ($\tau_{c,s}$) are included to eliminate any shocks common to all firms in the same sector and in the same country. The next section outlines the methodological framework for choosing among models.

## 4.2 Framework for choosing appropriate model

The framework proposed to guide researchers in their choice of a binary dependent variable model using SAFE draws on the econometrics literature regarding the LPM and non-linear models such as probit and logit. Fig 1 outlines the framework proposed. Put simply, it suggests that researchers should employ the LPM—given its ease of interpretation and computation and that its estimates are often reliable in practice—rather than probit or logit if four criteria are met. First, 100% of LPM probabilities should be bounded by 0–1 [21, 63, 83].

Second, strong correlations should exist between the LPM models and nonlinear models such as probit and logit [21]. Third, the estimates should be similar to those from nonlinear probability models [20–22, 63]. Fourth, the LPM model should perform relatively well compared to the probit and logit across a comprehensive range of goodness of fit statistics and test statistics. If the LPM does not meet these four criteria, the researcher should choose the best-performing model from the probit and logit models using the diagnostic framework outlined in Table 2.

These goodness of fit statistics proposed are outlined in Table 2 and include McFadden's pseudo $R^2$, McFaddens's adjusted pseudo $R^2$, percentage of correct predictions (PCP), percentage reduction in error (PRE), the expected PCP and PRE, the BIC and AIC and area under the ROC curve. McFadden's pseudo $R^2$ is chosen as this measure satisfies almost all of Kvålseth [41] eight criteria for a good $R^2$ [40, 41, 42]. The inference tests proposed are the Wald test and Likelihood ratio test. The Lagrange multiplier test is also used for joint significance of variables in the literature [44, 45]. The next section uses this proposed framework to choose among binary dependent variables and applies it to the model specification used by Finnegan and Kapoor [1].

## 5. Findings

This section reports the results using the most saturated empirical model (with macro, bank and firm controls with country sector and time fixed effects and robust heteroskedastic

standard errors) and sample over 2014–2019 from SAFE used by Finnegan and Kapoor [1] for the LPM, probit and logit model. It aims to assess which model, LPM, probit or logit is the most appropriate based on the methodological framework for evaluating models proposed in this paper. Section 5.1 presents the findings for H1: MP $_{t-2}$ x debt to assets increased. Section 5.2 presents the findings for H2: MP $_{t-2}$ x firm risk.

## 5.1. H1: MP$_{t-2}$ x Debt to assets increased

Fig 2 shows the quantile-quantile plot for probability of Debt to Assets increased with probit versus LPM. At first glance, the unboundedness so common in LPM is evident for this regression. On further investigation, Table 3 which shows the range of probabilities for H1 and the percentage of probabilities that fall within the range of 0 and 1.

It can be seen that 99.34% of observations fall inside the unit interval for this regression. If no (or very few) predicted probabilities lie outside the unit interval then the LPM is expected to be unbiased and consistent (or largely so) [21, 63]. Table 4 shows the correlation coefficients for H1: MP$_{t-2}$ x Debt to assets increased for logit, probit and LPM, and it can be seen that the correlations are over 0.995 and significant at the 1% level.

Table 5 shows H1: MP$_{t-2}$ x Debt to assets using the three econometrics methods. It can be seen that the estimates do not change for the main variable of interest (Further, estimates do not change for any of the control variables when estimated by the three different techniques and these more comprehensive results are reported in the S2–S6 Tables). The LPM is comparable in terms of PCP and PRE and the LPM is associated with a higher likelihood statistic. This regression meets the four criteria for choosing an LPM, and this may be preferred given its easier interpretation.

## 5.2. H2: MP$_{t-2}$ x firm risk

Firm risk is proxied using SAFE categorical variables: Profit decreased, credit history deteriorated, own outlook deteriorated, and own capital deteriorated and innovation (if the firm innovated in the previous six months). Fig 2 shows the quantile-quantile plots for probit versus LPM for all measures of risk, and it can be seen that in each case, the LMP is unbounded in the range of (0,1). However, Table 3 shows that the predicted probabilities fall outside the unit interval to a small degree for each case. Further, Table 4 shows the correlation coefficients for LPM, logit, and probit, which are above 0.99 and significant at the 1% level. Table 6 to 11 show the regressions for each interaction with various measures of risk: Profit decreased (Table 6), credit history deteriorated (Table 7), own outlook deteriorated (Table 8), own capital deteriorated (Table 9) and innovation (Table 10). Profit decreased, credit history deteriorated, outlook deteriorated, and capital deteriorated, and they do not meet all the criteria for an LPM. Consistent estimates across LPM and nonlinear models generally do not meet the criteria. In these cases, a nonlinear model is preferred. In each of these cases, the diagnostics indicate that logit and probit perform equally well across all goodness of fit statistics, but the probit model displays a higher Wald statistic and, therefore, is the most appropriate model for all regressions. MP$_{t-2}$ X Innovation does satisfy the criteria for choosing an LPM; the unit interval bounds 0.996% of its predicted probabilities, the correlation coefficients between LPM, logit and probit are high and significant, the estimates are the same for the three methodologies, the goodness of fit statistics are comparable, and the Likelihood ratio is higher for the LPM.

These findings illustrate a framework for choosing one binary dependent variable model over another. It is a valuable addition to the researcher's toolkit for taking a robust approach to research decisions on which model to employ in their analysis of SMEs credit constraints using SAFE data. The LPM is suitable for MP$_{t-2}$ x Debt to assets and MP$_{t-2}$ x Innovation when

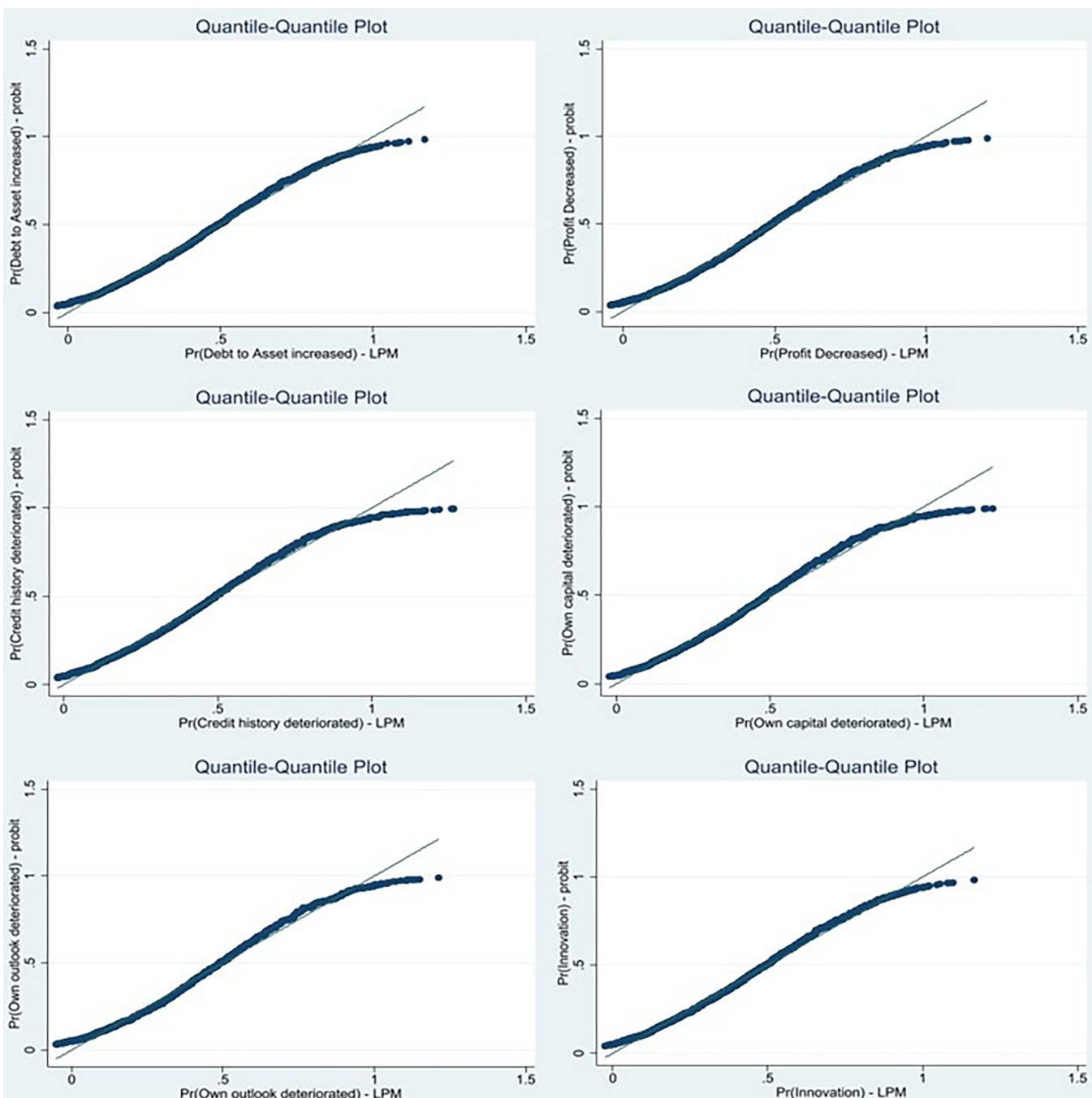

**Fig 2. Quantile-quantile plots for probit versus LPM, H1 and H2.** Source: Authors (2024).

applied to the SAFE sample and empirical model presented in Finnegan and Kapoor [1] as it satisfies the four criteria outlined in the diagnostic framework outlined in Fig 1. However, the probit model is more appropriate for $MP_{t-2}$ x profit decreased, $MP_{t-2}$ x credit history deteriorated, $MP_{t-2}$ x outlook deteriorated and $MP_{t-2}$ x capital deteriorated. Even though the LPM is suitable for $MP_{t-2}$ x Debt to assets and $MP_{t-2}$ x Innovation, if a researcher wishes to choose a probit model for consistency in their reporting, the probit model is more suitable given its

**Table 3. H1 and H2.** Range and percentage of probabilities 0–1 for LPM.

| | min | max | Percentage | Within range | Total obs |
|---|---|---|---|---|---|
| H1: $MP_{t-2}$ x Debt to assets | -0.0352 | 1.168813 | 0.99 | 8611 | 8668 |
| H2: $MP_{t-2}$ x Profit decreased | -0.0412 | 1.20213 | 0.983 | 8575 | 8726 |
| H2: $MP_{t-2}$ x Credit history deteriorated | -0.0202 | 1.265759 | 0.984 | 8639 | 8779 |
| H2: $MP_{t-2}$ x own outlook deteriorated | -0.0517 | 1.212226 | 0.962 | 8375 | 8707 |
| H2: $MP_{t-2}$ x own capital deteriorated | -0.0237 | 1.223208 | 0.979 | 8550 | 8734 |
| H2: $MP_{t-2}$ x Innovation | -0.02514 | 1.167221 | 0.996 | 8786 | 8820 |

higher Wald statistic indicating a better model fit. Further, there may be other reasons to choose a probit model. Logit estimators are on the log odds scale, whereas the probit models provide probabilities that are easier to interpret [1, 61]. Further, probit uses the normality assumption, allowing for easier analysis [20].

## 5.3 Multicollinearity and outliers

In the SAFE literature, there is barely any discussion on checks for multicollinearity and no discussion of outliers. In terms of multicollinearity, Sclip [11], Calabrese et al. [12] and Mac an Bhaird [25] (11.6% of studies) use the Variance Inflation Factor (VIF) while Mol-Gómez-Vázquez et al. [37] use correlation matrices to identify if there is correlation between the independent variables. There is no discussion of outliers, yet, SAFE is a cross-sectional survey conducted across EU countries, each experiencing different economic cycles over time. Yet, it is well known that either phenomenon can distort regression results, and this study checks to ensure that neither multicollinearity or outliers influence the results. In this study, correlation tables are used to identify pairwise correlations. The mean Variance Inflation Factor (VIF) is used to identify the extent to which a given explanatory variable can be explained by all the other explanatory variables in the equation.

The correlation tables in S7 Table indicates that, in general, pairwise correlation is not present to any great degree with no correlation in excess of 0.8, which is considered acceptable [19, 84, 85]. However, the correlation tables show perfect multicollinearity between the interaction term and its constituent parts. This is because there is structural multicollinearity in the model

**Table 4. Correlation coefficients between logit, probit and LPM.**

| H1 –Debt to Assets Increased | | | | H2—Profit decreased | | | |
|---|---|---|---|---|---|---|---|
| | Logit | Probit | LPM | | Logit | Probit | LPM |
| Logit | 1 | | | Logit | 1 | | |
| Probit | 1.000*** | 1 | | Probit | 1.000*** | 1 | |
| LPM | 0.996*** | 0.995*** | 1 | LPM | 0.995*** | 0.993*** | 1 |
| H2—Credit history deteriorated | | | | H2—Own outlook deteriorated | | | |
| | Logit | Probit | LPM | | Logit | Probit | LPM |
| Logit | 1 | | | Logit | 1 | | |
| Probit | 1.000*** | 1 | | Probit | 1.000*** | 1 | |
| LPM | 0.994*** | 0.991*** | 1 | LPM | 0.994*** | 0.991*** | 1 |
| H2—Own Capital Deteriorated | | | | H2—Innovation | | | |
| | Logit | Probit | LPM | | Logit | Probit | LPM |
| Logit | 1 | | | Logit | 1 | | |
| Probit | 1.000*** | 1 | | Probit | 1.000*** | 1 | |
| LPM | 0.994*** | 0.992*** | 1 | LPM | 0.997*** | 0.995*** | 1 |

**Table 5. H1: MP$_{t-2}$ x Debt to assets.**

| Credit constrained Variables | (1) LPM | (2) Probit | (3) Logit |
|---|---|---|---|
| MP$_{t-2}$ | -0.0326 | -0.0309 | -0.0316 |
|  | (0.030) | (0.029) | (0.028) |
| Debt to assets increased | -0.51*** | -0.46*** | -0.47*** |
|  | (0.143) | (0.130) | (0.129) |
| MP$_{t-2}$ x Debt to assets increased | 0.045*** | 0.042*** | 0.042*** |
|  | (0.012) | (0.011) | (0.011) |
| Observations | 8,668 | 8,668 | 8,668 |
| **Controls** |  |  |  |
| Country*Sector FE | Yes | Yes | Yes |
| Time FE | Yes | Yes | Yes |
| Bank Controls | Yes | Yes | Yes |
| Macro Controls | Yes | Yes | Yes |
| Firm Controls | Yes | Yes | Yes |
| **Goodness of Fit** |  |  |  |
| Mc Fadden's Pseudo R$^2$ |  | 0.18 | 0.181 |
| Mc Fadden's Adj Pseudo R$^2$ |  | 0.17 | 0.171 |
| PCP/Count R$^2$ | 0.730 | 0.732 | 0.732 |
| PRE /Adj Count R$^2$ | 0.277 | 0.282 | 0.283 |
| Expected PCP | - | 0.637 | 0.637 |
| Expected PRE | - | 0.223 | 0.225 |
| BIC | -68259.2 | -68706.8 | -68713.8 |
| AIC | 1.148 | 1.096 | 1.095 |
| ROC curve |  | 0.7672 | 0.7678 |
| Deviance | 9838.3 | 9390.6 | 9383.7 |
| **Multicollinearity** |  |  |  |
| Mean VIF | 21.95 | 21.92 | 21.92 |
| Mean VIF without interaction | 3.11 | 3.09 | 3.09 |
| **Outliers** |  |  |  |
| MP$_{t-2}$ x Debt to assets without obs. with dev residual > \|3\| | 0.0449*** | 0.04020**** | 0.04117*** |
| **Statistical inference** |  |  |  |
| LR | 2166.866 | 2016.236 | 1495.30 |
| Prob > LR | 0.0000 | 0.0000 | 0.0000 |
| Wald Chi2 $X^2$ |  | 1723.53 | 2068.177 |
| Prob > Chi$^2$ |  | 0.0000 | 0.0000 |
| F Test | 97.91 |  |  |
| $P = 0$ ($p$-value) | 0.000 |  |  |

The probability of being credit constrained is the dependent variable for stressed countries with LPM in column (1), probit in column (2) and logit in column (3). Marginal effects for probit and logit and average partial effects for LPM are reported. MP$_{t-2}$ is the one-year lag (equivalent to two survey waves) of the logarithm the assets of individual central bank balance sheets—minus autonomous factors—for stressed countries. Debt-to-assets increased is a categorical variable which equals 1 if the firm's debt-to-assets increased, and 0 if it remained the same or decreased in the previous six months. The hypothesis to be tested is that H1: MP$_{t-2}$ x Debt to assets increased is significant and negative. Country*sector and time fixed effects are included in all regressions. Robust standard errors are in the parentheses. ***, **, * represent significance at 1%, 5% and 10%, respectively.

as the key variable of interest for H1 and H2 is an interaction term: H1: $\gamma' DebttoAssets_{i,c,t} * MP_{c,t-2}$ and H2: $\gamma' FirmRisk_{i,c,t} * MP_{c,t-2}$. The Variance Inflator Factor (VIF) also reflects this structural multicollinearity. Tables 5–10 shows that when the interaction term

**Table 6. H2: $MP_{t-2}$ x profit decreased.**

| Credit constrained | (1) | (2) | (3) |
|---|---|---|---|
| Variables | LPM | Probit | Logit |
| $MP_{t-2}$ | -0.00537 | -0.00823 | -0.00917 |
| | (0.0304) | (0.0282) | (0.0278) |
| Profit decreased | 0.150 | 0.0657 | 0.0429 |
| | (0.137) | (0.120) | (0.117) |
| $MP_{t-2}$ x Profit decreased | -0.00112 | 0.00521 | 0.00707 |
| | (0.0114) | (0.0101) | (0.00988) |
| Observations | 8,726 | 8,726 | 8,726 |
| **Controls** | | | |
| Country*Sector FE | Yes | Yes | Yes |
| Time FE | Yes | Yes | Yes |
| Bank Controls | Yes | Yes | Yes |
| Macro Controls | Yes | Yes | Yes |
| Firm Controls | Yes | Yes | Yes |
| **Goodness of Fit** | | | |
| Mc Fadden's Pseudo $R^2$ | | 0.194 | 0.195 |
| Mc Fadden's Adj Pseudo $R^2$ | | 0.184 | 0.185 |
| PCP/Count $R^2$ | 0.736 | 0.737 | 0.737 |
| PRE /Adj Count $R^2$ | 0.294 | 0.296 | 0.298 |
| Expected PCP | | 0.644 | 0.645 |
| Expected PRE | | 0.239 | 0.241 |
| BIC | -68942.22 | -69381.4 | -69389.6 |
| AIC | 1.129 | 1.078 | 1.077 |
| ROC curve | | 0.7786 | 0.7791 |
| Deviance | 9738.971 | 9299.775 | 9291.57 |
| **Multicollinearity** | | | |
| Mean VIF | 23.33 | 21.98 | 21.98 |
| Mean VIF without interaction | 4.51 | 3.10 | 3.10 |
| **Outliers** | | | |
| $MP_{t-2}$ x credit history without obs. with dev residual > \|3\| | -0.00112 | 0.00570 | 0.0055 |
| **Statistical inference** | | | |
| LR | 2355.538 | 2237.365 | 2245.510 |
| Prob > LR | 0.0000 | 0.0000 | 0.00 |
| Wald Chi2 $X^2$ | | 1811.79 | 1568.64 |
| Prob > $Chi^2$ | | 0.0000 | 0.00 |
| F Test | 108.07 | | |
| $P = 0$ ($p$-value) | 0.0000 | | |

The probability of being credit constrained is the dependent variable for stressed countries with LPM in column (1), probit in column (2) and logit in column (3). Marginal effects for probit and logit and average partial effects for LPM are reported. $MP_{t-2}$ is the one-year lag (equivalent to two survey waves) of the logarithm the assets of individual central bank balance sheets—minus autonomous factors—for stressed countries. Profit decreased is a categorical variable which proxy firm risk from the firm's viewpoint. The hypothesis to be tested is that H2: $MP_{t-2}$ x Profit decreased is significant and negative. Bank controls (non-performing loans and tier-1 capital ratio) and macro controls (inflation and unemployment) are lagged by one-year (equivalent to two survey waves). Firm controls are added in column (2), (4) and (6). Country*sector and time fixed effects are included in all regressions. The omitted variable—which also serves as the reference category—for firm size is medium (50 to <250); the omitted firm turnover is above €50mn; the omitted sector is construction, and the omitted age is more than 10 years. Robust standard errors are in the parentheses. ***, **, * represent significance at 1%, 5% and 10%, respectively.

**Table 7. H2: $MP_{t-2}$ x credit history deteriorated.**

| Credit constrained | (2) | (4) | (6) |
|---|---|---|---|
| Variables | LPM | Probit | Logit |
| $MP_{t-2}$ | -0.0147 | -0.00923 | -0.00999 |
| | (0.0297) | (0.0274) | (0.0268) |
| Credit history deteriorated | -0.108 | 0.0990 | 0.154 |
| | (0.207) | (0.205) | (0.205) |
| $MP_{t-2}$ x Credit history deteriorated | 0.0308* | 0.0107 | 0.00641 |
| | (0.0175) | (0.0170) | (0.0170) |
| Observations | 8,779 | 8,779 | 8,779 |
| **Controls** | | | |
| Country*Sector FE | Yes | Yes | Yes |
| Time FE | Yes | Yes | Yes |
| Bank Controls | Yes | Yes | Yes |
| Macro Controls | Yes | Yes | Yes |
| Firm Controls | Yes | Yes | Yes |
| **Goodness of Fit** | | | |
| Mc Fadden's Pseudo $R^2$ | | 0.206 | 0.2067 |
| Mc Fadden's Adj Pseudo $R^2$ | | 0.197 | 0.197 |
| PCP/Count $R^2$ | 0.745 | 0.744 | 0.744 |
| PRE /Adj Count $R^2$ | 0.316 | 0.314 | 0.315 |
| Expected PCP | | 0.651 | 0.652 |
| Expected PRE | | 0.254 | 0.256 |
| BIC | -69587.5 | -70004.8 | -70011.88 |
| AIC | 1.109 | 1.062 | 1.061 |
| ROC curve | | 0.7882 | 0.7884 |
| Deviance | | 9210.157 | 9303.071 |
| **Multicollinearity** | | | |
| Mean VIF | 27.72 | 26.16 | 26.16 |
| Mean VIF without interaction | 4.73 | 3.08 | 3.08 |
| **Outliers** | | | |
| $MP_{t-2}$ x credit history without obs. with dev residual > \|3\| | 0.0300* | 0.01483 | 0.008 |
| **Statistical inference** | | | |
| LR | 2534.244 | 2391.042 | 2398.129 |
| Prob > LR | 0.0000 | 0.0000 | 0.00 |
| Wald Chi2 $X^2$ | | 1882.36 | 1646.48 |
| Prob > Chi$^2$ | | 0.0000 | 0.00 |
| F Test | 113.94 | | |
| $P = 0$ (*p*-value) | 0.0000 | | |

The probability of being credit constrained is the dependent variable for stressed countries with LPM in column (1), probit in column (2) and logit in column (3). Marginal effects for probit and logit and average partial effects for LPM are reported. $MP_{t-2}$ is the one-year lag (equivalent to two survey waves) of the logarithm the assets of individual central bank balance sheets—minus autonomous factors—for stressed countries. Credit history deteriorated is a categorical variable which proxy firm risk from the firm's viewpoint. The hypothesis to be tested is that H2: $MP_{t-2}$ x credit history deteriorated is significant and negative. Bank controls (non-performing loans and tier-1 capital ratio) and macro controls (inflation and unemployment) are lagged by one-year (equivalent to two survey waves). Firm controls are added in column (2), (4) and (6). Country*sector and time fixed effects are included in all regressions. The omitted variable—which also serves as the reference category—for firm size is medium (50 to <250); the omitted firm turnover is above €50mn; the omitted sector is construction, and the omitted age is more than 10 years. Robust standard errors are in the parentheses. ***, **, * represent significance at 1%, 5% and 10%, respectively.

**Table 8. H2: $MP_{t-2}$ x own outlook deteriorated.**

| Credit constrained | (1) | (2) | (3) |
|---|---|---|---|
| Variables | LPM | Probit | Logit |
| $MP_{t-2}$ | -0.00626 | -0.00648 | -0.00796 |
| | (0.0298) | (0.0278) | (0.0275) |
| Own outlook deteriorated | 0.196 | 0.114 | 0.0941 |
| | (0.152) | (0.135) | (0.133) |
| $MP_{t-2}$ x Own outlook deteriorated | 0.00378 | 0.00836 | 0.00972 |
| | (0.0127) | (0.0113) | (0.0112) |
| Observations | 8,707 | 8,707 | 8,707 |
| Country*Sector FE | Yes | Yes | Yes |
| Time FE | Yes | Yes | Yes |
| Bank Controls | Yes | Yes | Yes |
| Macro Controls | Yes | Yes | Yes |
| Firm Controls | Yes | Yes | Yes |
| **Goodness of Fit** | | | |
| Mc Fadden's Pseudo $R^2$ | | 0.217 | 0.218 |
| Mc Fadden's Adj Pseudo $R^2$ | | 0.208 | 0.209 |
| PCP/Count $R^2$ | 0.753 | 0.753 | 0.753 |
| PRE /Adj Count $R^2$ | 0.338 | 0.336 | 0.338 |
| Expected PCP | | 0.657 | 0.658 |
| Expected PRE | | 0.267 | 0.270 |
| BIC | -69305.31 | -69489.67 | -69500.148 |
| AIC | 1.094 | 1.046 | 1.045 |
| ROC curve | | 0.7966 | 0.7969 |
| Deviance | 9483.987 | 9000.257 | 8989.851 |
| **Multicollinearity** | | | |
| Mean VIF | 24.41 | 24.32 | 24.32 |
| Mean VIF without interaction | 3.19 | 3.10 | 3.10 |
| **Outliers** | | | |
| $MP_{t-2}$ x own outlook without obs. with dev residual > \|3\| | 0.00373 | 0.0099 | 0.0099 |
| **Statistical inference** | | | |
| LR | 2571.315 | 2499.446 | 2509.923 |
| Prob > LR | 0.0000 | 0.0000 | 0.00 |
| Wald Chi2 $X^2$ | | 1922.91 | 1660.99 |
| Prob > $Chi^2$ | | 0.0000 | 0.00 |
| F Test | 165.82 | | |
| $P = 0$ (p-value) | 0.0000 | | |

The probability of being credit constrained is the dependent variable for stressed countries with LPM in column (1), probit in column (2) and logit in column (3). Marginal effects for probit and logit and average partial effects for LPM are reported. $MP_{t-2}$ is the one-year lag (equivalent to two survey waves) of the logarithm the assets of individual central bank balance sheets—minus autonomous factors—for stressed countries. Own outlook deteriorated is a categorical variable which proxy firm risk from the firm's viewpoint. The hypothesis to be tested is that H2: $MP_{t-2}$ x Own outlook deteriorated is significant and negative. Bank controls (non-performing loans and tier-1 capital ratio) and macro controls (inflation and unemployment) are lagged by one-year (equivalent to two survey waves). Firm controls are added in column (2), (4) and (6). Country*sector and time fixed effects are included in all regressions. The omitted variable—which also serves as the reference category—for firm size is medium (50 to <250); the omitted firm turnover is above €50mn; the omitted sector is construction, and the omitted age is more than 10 years. Robust standard errors are in the parentheses. ***, **, * represent significance at 1%, 5% and 10%, respectively.

**Table 9. H2: MP$_{t-2}$ x own capital.**

| Credit constrained | (2) | (4) | (6) |
|---|---|---|---|
| variables | LPM | Probit | Logit |
| MP$_{t-2}$ | -0.0258 | -0.0171 | -0.0191 |
|  | (0.0298) | (0.0277) | (0.0273) |
| Own capital deteriorated | -0.184 | -0.0611 | -0.0290 |
|  | (0.188) | (0.169) | (0.168) |
| MP$_{t-2}$ x Profit decreased | 0.0353** | 0.0234 | 0.0204 |
|  | (0.0162) | (0.0143) | (0.0142) |
| Observations | 8,734 | 8,734 | 8,734 |
| **Controls** |  |  |  |
| Country*Sector FE | Yes | Yes | Yes |
| Time FE | Yes | Yes | Yes |
| Bank Controls | Yes | Yes | Yes |
| Macro Controls | Yes | Yes | Yes |
| Firm Controls | Yes | Yes | Yes |
| **Goodness of Fit** |  |  |  |
| Mc Fadden's Pseudo R$^2$ |  | 0.199 | 0.199 |
| Mc Fadden's Adj Pseudo R$^2$ |  | 0.189 | 0.190 |
| PCP/Count R$^2$ | 0.743 | 0.741 | 0.743 |
| PRE /Adj Count R$^2$ | 0.309 | 0.305 | 0.308 |
| Expected PCP |  | 0.647 | 0.648 |
| Expected PRE |  | 0.245 | 0.247 |
| BIC | -69096.81 | -69523.43 | -69528.43 |
| AIC | 1.119 | 1.070 | 1.070 |
| ROC curve |  | 0.7807 | 0.7809 |
| Deviance | 9664.935 | 9238.310 | 9233.308 |
| **Multicollinearity** |  |  |  |
| Mean VIF | 20.33 | 20.09 | 20.09 |
| Mean VIF without interaction | 3.34 | 3.09 | 3.09 |
| **Outliers** |  |  |  |
| MP$_{t-2}$ x Own Capital without obs. with dev residual > \|3\| | 0.0351** | 0.0232 | 0.0231 |
| **Statistical inference** |  |  |  |
| LR | 2423.316 | 2292.815 | 2297.816 |
| Prob > LR | 0.0000 | 0.0000 | 0.00 |
| Wald Chi2 $X^2$ |  | 1813.46 | 1573.64 |
| Prob > Chi$^2$ |  | 0.0000 | 0.00 |
| F Test | 114.11 |  |  |
| $P = 0$ (p-value) | 0.0000 |  |  |

The probability of being credit constrained is the dependent variable for stressed countries with LPM in column (1), probit in column (2) and logit in column (3). Marginal effects for probit and logit and average partial effects for LPM are reported. MP$_{t-2}$ is the one-year lag (equivalent to two survey waves) of the logarithm the assets of individual central bank balance sheets—minus autonomous factors—for stressed countries. Own capital deteriorated is a categorical variable which proxy firm risk from the firm's viewpoint. The hypothesis to be tested is that H2: MP$_{t-2}$ x Own capital deteriorated is significant and negative. Bank controls (non-performing loans and tier-1 capital ratio) and macro controls (inflation and unemployment) are lagged by one-year (equivalent to two survey waves). Firm controls are added in column (2), (4) and (6). Country*sector and time fixed effects are included in all regressions. The omitted variable—which also serves as the reference category—for firm size is medium (50 to <250); the omitted firm turnover is above €50mn; the omitted sector is construction, and the omitted age is more than 10 years. Robust standard errors are in the parentheses. ***, **, * represent significance at 1%, 5% and 10%, respectively.

**Table 10. H2: MP$_{t-2}$ x Innovation.**

| Credit constrained | (2) | (4) | (6) |
|---|---|---|---|
| Variables | LPM | Probit | Logit |
| MP$_{t-2}$ | -0.0176 | -0.0169 | -0.0170 |
| | (0.0302) | (0.0283) | (0.0280) |
| Innovation | -0.101 | -0.0960 | -0.0810 |
| | (0.135) | (0.122) | (0.120) |
| MP$_{t-2}$ x Innovation | 0.00984 | 0.00955 | 0.00829 |
| | (0.0113) | (0.0103) | (0.0101) |
| Observations | 8,820 | 8,820 | 8,820 |
| Country*Sector FE | Yes | Yes | Yes |
| Time FE | Yes | Yes | Yes |
| Bank Controls | Yes | Yes | Yes |
| Macro Controls | Yes | Yes | Yes |
| Firm Controls | Yes | Yes | Yes |
| **Goodness of Fit** | | | |
| Mc Fadden's Pseudo R$^2$ | | 0.177 | 0.178 |
| Mc Fadden's Adj Pseudo R$^2$ | | 0.168 | 0.168 |
| PCP/Count R$^2$ | 0.728 | 0.732 | 0.732 |
| PRE /Adj Count R$^2$ | 0.272 | 0.281 | 0.281 |
| Expected PCP | | 0.635 | 0.636 |
| Expected PRE | | 0.220 | 0.222 |
| BIC | -69586.162 | -7000.456 | -70047.971 |
| BIC' | -1818.678 | -1711.026 | -1717.541 |
| AIC | 1.151 | 1.099 | 1.099 |
| AIC*n | 10151.910 | 9696.616 | 9690.101 |
| ROC curve | | 0.7642 | 0.7649 |
| Deviance | 10041.910 | 9586.616 | 9580.101 |
| **Multicollinearity** | | | |
| Mean VIF | 22.13 | 20.78 | 20.78 |
| Mean VIF without interaction | 4.52 | 3.09 | 3.09 |
| **Outliers** | | | |
| MP$_{t-2}$ x Innovation without obs. with dev residual > \|3\| | 0.0098 | 0.009633 | 0.0081 |
| **Statistical inference** | | | |
| LR | 2172.985 | 2065.332 | 1508.64 |
| Prob > LR | 0.0000 | 0.0000 | 0.00 |
| Wald Chi2 $X^2$ | | 1739.07 | 1322.842 |
| Prob > Chi$^2$ | | 0.0000 | 0.00 |
| F Test | 98.03 | | |
| $P = 0$ (*p*-value) | 0.0000 | | |

The probability of being credit constrained is the dependent variable for stressed countries with LPM in column (1), probit in column (2) and logit in column (3). Marginal effects for probit and logit and average partial effects for LPM are reported. MP$_{t-2}$ is the one-year lag (equivalent to two survey waves) of the logarithm the assets of individual central bank balance sheets—minus autonomous factors—for stressed countries. Innovation is a categorical variable which proxies if the firm innovated in the previous six months and is a measure of firm risk. The hypothesis to be tested is that H2: MP$_{t-2}$ x Innovation is significant and negative Bank controls (non-performing loans and tier-1 capital ratio) and macro controls (inflation and unemployment) are lagged by one-year (equivalent to two survey waves). Firm controls are added in column (2), (4) and (6). Country*sector and time fixed effects are included in all regressions. The omitted variable—which also serves as the reference category—for firm size is medium (50 to <250); the omitted firm turnover is above €50mn; the omitted sector is construction, and the omitted age is more than 10 years. Robust standard errors are in the parentheses. ***, **, * represent significance at 1%, 5% and 10%, respectively.

is present the VIF is between 22 and 28 (Tables 5–10). However, when the interaction term is excluded, the VIF falls below 5, which is considered to indicate an acceptable level of multicollinearity [53, 86, 87].

One solution to interaction terms and structural multicollinearity cited in the literature is to mean centre a constituent variable (standardise the variable by subtracting the mean) [27, 53, 87]. However, this is not possible with a binary variable like *DebttoAssets* or *FirmRisk*. Another solution is to increase the sample size as logit and probit regression uses maximum likelihood estimation (MLE), which relies on large-sample asymptotic normality. This means that the reliability of estimates increases when the sample size is large enough [17, 18, 88]. However, the sample size (11,319, which drops to 8,777 on account of the monetary policy lag employed) cannot be increased as it is determined by the research focus on stressed countries and credit-constrained SMEs from 2014–2019.

This study acknowledges and underlines the necessary degree of multicollinearity arising from the interaction term due to its importance in addressing the hypothesis. This is justified because standard errors are relatively small, and the estimates do not change much when more variables are added to the model (S2–S6 Tables), indicating that multicollinearity does not compromise the model [19, 61, 89].

In terms of outliers, there is no discussion in the SAFE literature, despite SAFE being a cross-sectional survey across EU countries with different countries experiencing different economic cycles over time. A single observation that is substantially different from all other observations can make a significant difference in the regression analysis results [19, 53, 90]. Studies using SAFE need to consider outliers explicitly, and this paper identifies outliers for each regression and runs the regressions, excluding the outliers to identify if the estimates change. In particular, this study takes the following three steps to identify if the outliers distort the estimates:

1. Identifies the number of outliers—observations with deviance residuals $> |3|$ and their average [91, 92]. Residuals greater than the absolute value of 3 are in the tails of a standard normal distribution and this usually indicates strain in the model [91]. Deviance residuals can be roughly approximated with a standard normal distribution when the model holds [92].

2. Plots the deviance residuals against the predicted values for credit constrained to identify any extreme values

3. Run the regressions without the observations with deviance residual $> |3|$

In terms of outliers, Tables 5–10 show the estimates for the interaction terms for both H1 and H2 when observations with a deviance residual $> |3|$ are excluded; the main results are very similar, which ensures confidence in the results [91]. The next section concludes the research study.

## 6. Conclusion

SMEs' access to bank finance has been a significant focus for researchers given that they are critical for the euro area economy and they are primarily bank dependent. SAFE data, especially responses to Q7a and Q7b, which document outcomes of SMEs' application for bank loans, has been employed extensively in the literature to study SME access to bank finance. Most of these studies construct a categorical dependent variable from SAFE data, with probit emerging as the dominant model (57%), followed by logit (28%) and then LPM (7.4%). However, the literature fails to provide a justification for their methodological approach; simply

citing a categorical dependent model requires a nonlinear model without considering the superior simplicity of interpreting LPM. Further, there is a dearth of goodness of fit and inference test statistics employed in the literature to determine if the models are a good fit or if they are statistically significant. There is thus a need to provide insights into analysis using SAFE data and binary dependent models. This research offers an initial contribution to understanding an appropriate methodological framework in the context of SAFE data and categorical dependent variables. It adds to the literature by providing a framework for researchers to choose one model over the other by identifying four criteria for the LPM to be considered an appropriate model. Further, it provides a diagnostic framework that researchers can deploy to ensure that their models perform robustly. The research study uses the sample and empirical model presented in Finnegan and Kapoor [1] to apply this framework and finds that their probit model was appropriate in all cases. However, this analysis shows that Finnegan and Kapoor [1] could have used a LPM in two cases: when considering the interaction between monetary policy and debt to assets and monetary policy and innovation. The use of the LPM is justified as a less complex econometric model, allowing for clearer communication of the results. By proposing a robust methodological framework when using SAFE to investigate SMEs' access to bank finance, this paper fosters further robust research using the SAFE dataset to investigate SMEs' access to finance within Europe by adding to the researcher's toolkit in the practical and robust application of methodologies.

Some limitations to this study require further research to keep building a solid framework on using SAFE data for policy making. This study employs robust standard errors but does not investigate the rationale for this standard error treatment. Other standard error treatments, such as clustering at the country level, may be more appropriate as these regressions combine the aggregate effect of monetary policy on micro units by merging aggregate data with micro-observations from SAFE [93]. Further, this study does not employ firm fixed effects—in line with the majority of literature which uses SAFE—given the cross-sectional nature of the SAFE firm data. Using firm fixed effects would further isolate the impact of UMP on SME bank liquidity limitations by absorbing any firm-specific credit demand shocks. The appropriate use of standard errors and firm fixed effects is worthy of further investigation.

## Supporting information

**S1 Data.**
(CSV)

**S1 Table. Variables, definition and data source.**
(DOCX)

**S2 Table. H1, probit versus LPM.**
(DOCX)

**S3 Table. H1, probit versus logit.**
(DOCX)

**S4 Table. H2, LPM.**
(DOCX)

**S5 Table. H2, probit.**
(DOCX)

**S6 Table. H2, logit.**
(DOCX)

**S7 Table. Correlation tables H1 and H2.**
(DOCX)

# Author Contributions

**Conceptualization:** Marie Finnegan.

**Data curation:** Marie Finnegan.

**Formal analysis:** Marie Finnegan.

**Investigation:** Marie Finnegan.

**Methodology:** Marie Finnegan.

**Project administration:** Marie Finnegan.

**Supervision:** Lucía Morales.

**Validation:** Marie Finnegan.

**Writing – original draft:** Marie Finnegan.

**Writing – review & editing:** Marie Finnegan.

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
