## [Decision Letter · Decision Letter 0]

15 Apr 2024

PONE-D-24-10511A methodological framework for exploring SME finance access with SAFE dataPLOS ONE

Dear Dr. Finnegan,

Thank you for submitting your manuscript to PLOS ONE. After careful consideration, we feel that it has merit but does not fully meet PLOS ONE’s publication criteria as it currently stands. Therefore, we invite you to submit a revised version of the manuscript that addresses the points raised during the review process.

We look forward to receiving your revised manuscript.

Kind regards,

Daphne Nicolitsas

Academic Editor

PLOS ONE

Journal Requirements:

Reviewers' comments:

Reviewer's Responses to Questions

**Comments to the Author**

1. Is the manuscript technically sound, and do the data support the conclusions?

Reviewer #1: Yes

Reviewer #2: Yes

2. Has the statistical analysis been performed appropriately and rigorously? 

Reviewer #1: Yes

Reviewer #2: No

3. Have the authors made all data underlying the findings in their manuscript fully available?

Reviewer #1: Yes

Reviewer #2: Yes

4. Is the manuscript presented in an intelligible fashion and written in standard English?

Reviewer #1: Yes

Reviewer #2: No

5. Review Comments to the Author

Reviewer #1: Dear Authors,

The study titled "A methodological framework for exploring SME finance access with SAFE data." is an overall successful study. However, there are some shortcomings. I strongly recommend you to make the following revisions.

Please write your own hypotheses and refer to the literature. Also express the hypotheses verbally. Write clearly how this study differs from the literature. Write the table names formally. You can add hypothesis tests as notes under the table. Write the limitations of the study and suggestions for future studies. There are frequent references to Finnegan and Kapoor in the study. Is this an extension study? If so, state this in the study, if not, state this clearly in the literature section and state that you have followed this study in other literature. This is a blind review, so it is seen as an extension since the identity of the authors is known and this study is also the authors.

All the best,

Reviewer #2: In my opinion, the paper offers a good contribution. So, I recommend accepting this paper, but after making the following modifications:

1- I think that the abstract needs improvement.

2- Some equations need correction. Moreover, in some equations, there are some symbols that are not defined.

3- Authors should test multicollinearity and outliers in the data before estimating the model. This point is very important for this study. Furthermore, adding references addressed the two problems in the LPM, logit, and probit models.

4- The authors should put the dataset used of this study in the “Supporting Information File".

5- At the conclusion of this work, limitations of this research should be mentioned.

6. PLOS authors have the option to publish the peer review history of their article (what does this mean?). If published, this will include your full peer review and any attached files.

Reviewer #1: No

Reviewer #2: **Yes: **‪Mohamed R. Abonazel‬‏

---

## [Author Response · Author response to Decision Letter 0]

3 Jun 2024

Response to Reviewers

Dear Reviewers,

Thank you very much for your insightful and valuable comments regarding this research paper. We have carefully addressed each comment and include a detailed response to each one below.

Reviewer#1

Dear Reviewer #1, thank you very much for your insightful comments. The research paper has been subject to a comprehensive review to ensure that all your comments are addressed. Your suggestions and recommendations makes the paper more readable and stronger. We respond to each comment in detail below:

 Please write your own hypotheses and refer to the literature. Also express the hypotheses verbally. 

We have updated section 4.1 Empirical model to write the hypothesis verbally and refer to the literature as follows:

For H1, γ^' is an interaction term and is the main coefficient of interest that captures if firms with increased debt-to-assets decreased their probability of being credit-constrained during UMP. It is expected that UMP for leveraged firms should make accessing bank finance easier via their improved balance sheets and collateral, and this should translate into a reduction in credit constraints (80, 81). A negative relationship is, therefore, expected between the probability that a firm is credit-constrained and the interaction between increased debt-to-assets ratio and UMP. 

For H2, the interaction between monetary policy and firm risk variables is the main coefficient of interest - γ^' - as it captures the probability of a risky firm being credit-constrained during periods of UMP. 〖FirmRisk〗_(i,c,t) is modelled categorically, and the measures for firm risk include a future predictor of risk - profit decreased in the previous six months, as well as a selection of subjective measures of risk - the firm's own view if there has been deterioration in credit history, own economic outlook and own capital in the previous six months and finally, an activity-based measure of risk - innovative activity given that such activity is more uncertain and therefore riskier. Given the low-interest rate environment generated by UMP, banks are expected to chase higher yields. This will manifest in lending to riskier firms via the monetary policy risk-taking channel (76, 82). A negative relationship is, therefore, expected between the probability that a firm is credit-constrained and the interaction between increased firm risk and UMP. 

 Write clearly how this study differs from the literature. 

We added the following to the introduction

This paper contributes to existing literature by introducing a methodological framework aimed at aiding researchers in determining the preferable categorical dependent model when utilising SAFE data. 

 Write the table names formally. You can add hypothesis tests as notes under the table. 

We have rewritten all the table names formally. We have added a hypothesis test note under each table.

For example: Table 5. The hypothesis being tested is that H1: MPt-2 x Debt to assets increased is significant and negative

 Write the limitations of the study and suggestions for future studies. 

We have added this section to the conclusion:

Some limitations to this study require further research to keep building a solid framework on using SAFE data for policy making. This study employs robust standard errors but does not investigate the rationale for this standard error treatment. Other standard error treatments, such as clustering at the country level, may be more appropriate as these regressions combine the aggregate effect of monetary policy on micro units by merging aggregate data with micro-observations from SAFE (93). Further, this study does not employ firm fixed effects - in line with the majority of literature which uses SAFE - given the cross-sectional nature of the SAFE firm data. Using firm fixed effects would further isolate the impact of UMP on SME bank liquidity limitations by absorbing any firm-specific credit demand shocks. The appropriate use of standard errors and firm fixed effects is worthy of further investigation. 

 There are frequent references to Finnegan and Kapoor in the study. Is this an extension study? If so, state this in the study, if not, state this clearly in the literature section and state that you have followed this study in other literature. This is a blind review, so it is seen as an extension since the identity of the authors is known and this study is also the authors.

Yes, it is an extension study of Finnegan and Kapoor (2023) and this is now stated clearly in the introduction and methodology sections. The research study builds on the mentioned research paper as it provides a good framework for data analysis and testing of outlined hypotheses.

Introduction: It is an extension study on the recent work by Finnegan and Kapoor (1), which compares their probit model to an LPM and logit model.

Methodology: . In particular, it in an extension of the study by Finnegan and Kapoor (1) - which studies the impact of unconventional monetary policy on SMEs' access to bank finance - in terms of assessing the methodological framework when using binary dependent variables and SAFE

Reviewer #2: In my opinion, the paper offers a good contribution. So, I recommend accepting this paper, but after making the following modifications:

Dear Reviewer #2, thank you very much for your valuable comments which we have taken on board to strengthen the paper. We address each one in detail below.

1- I think that the abstract needs improvement.

We have expanded the abstract to describe the main objective of the study and summarise the most important results and their significance. The revised abstract is as follows:

Abstract

Research on small and medium-sized enterprises (SMEs) access to bank finance is vital for the euro area economy. SMEs heavily represent the European business sector, employing around 100 million people and accounting for more than half of Gross Domestic Product. Research studies in the field often rely on the ECB/EC Survey on the Access to Finance of Enterprises (SAFE). Many studies employ probit or logit models with categorical dependent variables derived from SAFE. The research findings show that hardly any study employs the simpler linear probability model (LPM), with a dominant lack of research providing evidence that justifies the model selection process and suitability. However, it is well known that different econometrics models can lack consistency and frequently yield different results. Yet, the literature has no consensus on the best econometric approach. In addition, there is a lack of robustness tests in the literature to ensure model validity, underlining the need for a comprehensive review of the methodological framework that dominates SAFE data use. This paper addresses the identified research gap by introducing a robust methodological framework that helps researchers identify and choose an appropriate categorical model when using SAFE data. The study adds significant value to the extant literature by identifying four criteria that need to be considered when selecting the appropriate model among three common binary dependent models: LPM, probit and logit models. The findings show that the probit model was appropriate is all cases but that the LPM should not be disregarded, as it can be used in two cases: when considering the interaction between monetary policy and debt to assets and monetary policy and innovation. The use of the LPM is justified as a less complex econometric model, allowing for clearer communication of the results. This innovative, robust approach to choosing the appropriate econometric categorical dependent model when employing SAFE data contributes to support policy effectively.

2- Some equations need correction. Moreover, in some equations, there are some symbols that are not defined.

Hypothesis 1 (H1): UMP decreases the probability of firms with increased debt-to-assets being credit constrained.

To test H1, they model Equation 1 as follows:

P(〖CreditConstrained〗_(i,c,t)=1│x)=α_i+β^' 〖MP〗_(c,t-2)+γ^' 〖DebttoAssets〗_(i,c,t)*〖MP〗_(c,t-2)+δ^' X_(i,c,t)+θ^' 〖Macro〗_(c,t-2)+φ^' 〖BankCh〗_(c,t-2)+τ_(c,s)+ε_(i,s,c,t) 

Hypothesis 2 (H2): UMP reduces the probability of risky firms being credit constrained

To test H2, they model Equation 2 as follows:

P(〖CreditConstrained〗_(i,c,t)=1│x)=α_i+β^' 〖MP〗_(c,t-2)+γ^' 〖FrimRisk〗_(i,c,t)*〖MP〗_(c,t-2)+δ^' X_(i,c,t)+θ^' 〖Macro〗_(c,t-2)+φ^' 〖BankCh〗_(c,t-2)+τ_(c,s)+ε_(i,s,c,t), 

We include table S1 to ensure that all symbols in the equation are defined.

Variable Data Source Definition

Dependent variable

P(〖CreditConstrained〗_(i,c,t)=1│x) ECB/EC Survey on the access to finance of enterprises (SAFE)

Q7a, Q7b Binary variable = 1 if the firm reported (i) to have applied for bank loans in the previous six months but was rejected (Credit Denied) or (ii) to have applied but received less than 75% of its demand (Rationed) or (iii) to have refused credit because it was offered at a too high cost (Refused due to high cost) or (iv) not to have applied because of possible rejection (Discouraged). 0 = if the firm reported (i) to have applied for bank loans in the previous six months and they received everything or (ii) they received 75 percent or more of their demand. 

Independent variable H1

H1 γ^' 〖DebttoAssets〗_(i,c,t)*〖MP〗_(c,t-2) 

Debt to Assets increased ECB/EC SAFE Q2 = 1 if the firm’s debt to assets increased in the past 6 months, 0 if it remained unchanged or decreased

MPc,t-2 = ECB BS Assets (million) ECB Statistical Data Warehouse

 Continuous variable, monthly, total ECB assets (after subtracting the autonomous factors that are beyond the direct control of the ECB including banknotes in circulation and government balances) from individual central bank balance sheet for stressed countries, following Peydro et al. (2021). Monthly data averaged over half years ending in March and September.

Independent variable H2

H2 〖MP〗_(c,t-2)+γ^' 〖FirmRisk〗_(i,c,t) 

Where 〖FirmRisk〗_(i,c,t)= 

Profit decreased ECB/EC SAFE Q2 = 1 if the firm's profit decreased in the past 6 months, 0 if it remained unchanged or increased

Credit history deteriorated ECB/EC SAFE Q11 = 1 if the firm's credit history deteriorated in the previous 6 months, 0 if it remained the unchanged or improved

Own capital deteriorated ECB/EC SAFE Q11 = 1 if the firm's own capital deteriorated in the previous 6 months, 0 if it remained unchanged or improved

Own outlook deteriorated ECB/EC SAFE Q11 = 1 if the firm's own outlook deteriorated in the previous 6 months, 0 if it remained unchanged or improved

Innovation ECB/EC SAFE Q1 = 1 if the innovated (in terms of new or improved product, new or improved production process, new organisation of management, new way of selling goods or services) in the previous 6 months, 0 if did not

X_(i,c,t) = Firm controls 

Micro ECB/EC SAFE

QD1 = 1 if the firm has between 1 and 9 employees, 0 otherwise

Micro ECB/EC SAFE

QD1 = 1 if the firm has between 1 and 9 employees, 0 otherwise

Small ECB/EC SAFE

QD1 = 1 if the firm has between 10 and 49 employees, 0 otherwise

Medium ECB/EC SAFE

QD1 = 1 if the firm has between 50 and 249 employees, 0 otherwise

More than 10 years ECB/EC SAFE

QD5 = 1 if the firm is 10+ years old, 0 otherwise

Between 5 and 10 yrs ECB/EC SAFE

QD5 = 1 if the firm is between 5 and 10 years old, 0 otherwise

Between 2 and 5 yrs ECB/EC SAFE

QD5 = 1 if the firm is between 2 and 5 years old, 0 otherwise

Less than 2 yrs ECB/EC SAFE

QD5 = 1 if the firm is less than 2 years old, 0 otherwise

Stand-alone firm ECB/EC SAFE

QD2 = 1 if the firm is an autonomous profit-oriented enterprise, 0 otherwise

Individual or family owned ECB/EC SAFE

QD2 = 1 if the firm's owner is an individual or a family, 0 otherwise

Turnover up to 2mn ECB/EC SAFE

QD4 = 1 if the firm's annual turnover is less than €2 mln, 0 otherwise

Turnover between 2 and 10mn ECB/EC SAFE

QD4 = 1 if the firm's annual turnover is between €2 mln. and €10 mln, 0 otherwise

Turnover between 10 and 50mn ECB/EC SAFE

QD4 = 1 if the firm's annual turnover is between €10 mn. and €50 mn, 0 otherwise

Industry ECB/EC SAFE

QD3 = 1 if the firm's main activity is in industry, 0 otherwise

Construction ECB/EC SAFE

QD3 = 1 if the firm's main activity is in construction, 0 otherwise

Wholesale or Retail Trade ECB/EC SAFE

QD3 = 1 if the firm's main activity is in wholesale or retail trade, 0 otherwise

Services ECB/EC SAFE

QD3 = 1 if the firm's main activity is services, 0 otherwise

〖Macro〗_(c,t-2) = Macroeconomic controls 

Unemployment Eurostat Continuous variable, unemployment rate (share of active population), seasonally adjusted. The unemployment rate is the number of people unemployed expressed as a share of the labour force. The labour force is the total number of people employed and unemployed. Quarterly data averaged over half years ending in March and September and expressed as decimals.

Inflation Eurostat Continuous variable, inflation rate measured by HICP monthly data (annual rate of change). Average of monthly inflation rate data over half years ending in March and September and expressed as decimals.

〖BankCh〗_(c,t-2)= Bank controls 

Non-performing loans IMF Financial Soundness Indicators

 Continuous variable, quarterly non-performing loans as a share of total gross loans averaged over half years ending in March and September and expressed as decimals.

Regulatory capital to risk-weighted assets ratio IMF Financial Soundness Indicators

 Continuous variable, quarterly regulatory tier 1 capital as a share of risk weighted assets averaged over half years ending in March and September and expressed as decimals.

OR

We also clarify these definitions in the text of section 4.1 empirical model to define each symbol in the definition. See changes below:

This dependent variable ‘credit constrained’ equals 1 if the firm reported to have (i) applied for bank loans in the previous six months but was rejected (Credit Denied) or (ii) applied but received less than 75% of its demand (Rationed) or (iii) refused credit because it was offered at too high a cost (Refused due to high cost) or (iv) not applied because of possible rejection (Discouraged). Alternatively, the variable equals 0 if the firm reported to have applied for bank loans in the previous six months and received everything or received 75% and above. Given that the indicator is equal to 1 if the firm is credit constrained, a negative coefficient indicates that SMEs are less likely to be credit constrained.

MPt−2 is the one-year lag (equivalent to two survey waves in SAFE) of the logarithm the assets of individual central bank balance sheets - minus autonomous factors - for stressed countries. 

H1. Debt-to-assets increased is a categorical variable which equals 1 if the firm’s debt-to-assets increased, and 0 if it remained the same or decreased in the previous six months.

H2. 〖FirmRisk〗_(i,c,t) is modelled categorically and the measures for firm risk include a future predictor of risk - profit decreased in the previous six months, as well as a selection of subjective measures of risk - firms own view if there has been deterioration in credit history, own economic outlook and own capital in the previous six months and finally, an activity-based measure of risk - innovative activity given that such activity is more uncertain and therefore riskier.

3- Authors should test multicollinearity and outliers in the data before estimating the model. This point is very important for this study. Furthermore, adding references addressed the two problems in the LPM, logit, and probit models.

Thank you very much for this comment. We agree that this is very important for validity of results. We have therefore added a new section 5.3 on ‘multicollinearity and outliers’ to address this concern. We have also updated Tables 5-10 with measures of multicollinearity and results if outliers are excluded. New section below:

5.3 Multicollinearity and outliers

In the SAFE literature, there is barely any discussion on checks for multicollinearity and no discussion of outliers. In terms of multicollinearity, Mac an Bhaird (25), Calabrese et al. (12) a

---

## [Decision Letter · Decision Letter 1]

4 Jul 2024

A methodological framework for exploring SME finance with SAFE data

PONE-D-24-10511R1

Dear Dr. Finnegan,

We’re pleased to inform you that your manuscript has been judged scientifically suitable for publication and will be formally accepted for publication once it meets all outstanding technical requirements.

Kind regards,

Daphne Nicolitsas

Academic Editor

PLOS ONE

Additional Editor Comments (optional):

Reviewers' comments:

Reviewer's Responses to Questions

**Comments to the Author**

1. If the authors have adequately addressed your comments raised in a previous round of review and you feel that this manuscript is now acceptable for publication, you may indicate that here to bypass the “Comments to the Author” section, enter your conflict of interest statement in the “Confidential to Editor” section, and submit your "Accept" recommendation.

Reviewer #1: All comments have been addressed

2. Is the manuscript technically sound, and do the data support the conclusions?

Reviewer #1: Yes

3. Has the statistical analysis been performed appropriately and rigorously? 

Reviewer #1: Yes

4. Have the authors made all data underlying the findings in their manuscript fully available?

Reviewer #1: Yes

5. Is the manuscript presented in an intelligible fashion and written in standard English?

Reviewer #1: Yes

6. Review Comments to the Author

Reviewer #1: The authors have revised the points I had highlighted as important in the previous review round. All comments are addressed. Thank you.

7. PLOS authors have the option to publish the peer review history of their article (what does this mean?). If published, this will include your full peer review and any attached files.

Reviewer #1: No

---

## [Editor Report · Acceptance letter]

10 Jul 2024

PONE-D-24-10511R1 

PLOS ONE

Dear Dr. Finnegan, 

I'm pleased to inform you that your manuscript has been deemed suitable for publication in PLOS ONE. Congratulations! Your manuscript is now being handed over to our production team.

Kind regards, 

on behalf of

Dr. Daphne Nicolitsas 

Academic Editor

PLOS ONE